# Coapplication of Magnesium Supplementation and Vibration Modulate Macrophage Polarization to Attenuate Sarcopenic Muscle Atrophy through PI3K/Akt/mTOR Signaling Pathway

**DOI:** 10.3390/ijms232112944

**Published:** 2022-10-26

**Authors:** Can Cui, Zhengyuan Bao, Simon Kwoon-Ho Chow, Ronald Man Yeung Wong, Ailsa Welch, Ling Qin, Wing Hoi Cheung

**Affiliations:** 1Musculoskeletal Research Laboratory, Department of Orthopaedics and Traumatology, Hong Kong Special Administrative Region, The Chinese University of Hong Kong, Hong Kong 999077, China; 2Norwich Medical School, University of East Anglia, Norwich Research Park, Norwich NR4 7TJ, UK

**Keywords:** magnesium, LMHFV, sarcopenia, macrophage, atrophy, PI3K/Akt/mTOR pathway

## Abstract

Sarcopenia is an age-related geriatric syndrome characterized by the gradual loss of muscle mass and function. Low-magnitude high-frequency vibration (LMHFV) was shown to be beneficial to structural and functional outcomes of skeletal muscles, while magnesium (Mg) is a cofactor associated with better indices of skeletal muscle mass and strength. We hypothesized that LMHFV, Mg and their combinations could suppress inflammation and sarcopenic atrophy, promote myogenesis via PI3k/Akt/mTOR pathway in senescence-accelerated mouse P8 (SAMP8) mice and C2C12 myoblasts. Results showed that Mg treatment and LMHFV could significantly decrease inflammatory expression (C/EBPα and LYVE1) and modulate a CD206-positive M2 macrophage population at month four. Mg treatment also showed significant inhibitory effects on FOXO3, MuRF1 and MAFbx mRNA expression. Coapplication showed a synergistic effect on suppression of type I fiber atrophy, with significantly higher IGF-1, MyoD, MyoG mRNA (*p* < 0.05) and pAkt protein expression (*p* < 0.0001) during sarcopenia. In vitro inhibition of PI3K/Akt and mTOR abolished the enhancement effects on myotube formation and inhibited MRF mRNA and p85, Akt, pAkt and mTOR protein expressions. The present study demonstrated that the PI3K/Akt/mTOR pathway is the predominant regulatory mechanism through which LMHFV and Mg enhanced muscle regeneration and suppressed atrogene upregulation.

## 1. Introduction

Sarcopenia is an age-related geriatric syndrome associated with subsequent disability and morbidity including falls, fractures, physical disability and eventually mortality [1], with the diagnostic algorithm first defined by the European Working Group on Sarcopenia in Older People (EWGSOP) using muscle strength, mass and physical performance. Sarcopenia is increasingly recognized not only as an age-related problem but also one associated with a range of long-term conditions. Sarcopenia is considered ‘primary’ (or age-related) when no other specific cause is evident, whereas it is considered ‘secondary’ when causal factors other than (or in addition to) ageing are evident [2]. Several factors contribute to the development of sarcopenia that directly affect muscle tissues, including mitochondrial dysfunction, apoptosis, neural plaque changes and ageing-related motor neuron loss, satellite cell dysfunction, imbalance in protein metabolism, lack of physical activity, changes in endocrine function and inflammatory processes. As muscle mass is regulated by a dynamic balance of anabolic and catabolic processes, the myogenic regulatory factor (MRF) family of transcription factors, Myf5, Myf6, MyoD and MyoG (Myogenin), play a crucial role in directing satellite cell functions to regenerate mature skeletal muscle [3,4]. Recent findings showed that activation of the PI3K/Akt/mTOR pathway was important as a response to mechanical stimulation in muscle hypertrophy [5]. Reduction of Akt phosphorylation caused by a decrease of insulin-like growth factor (IGF-1) in older people could activate FOXO3, promoting the transcription of the atrogenes MAFbx (muscle atrophy F-box) and MuRF1 (muscle ring-finger protein 1), leading to decrease in muscle mass and fiber size [6,7,8].

Skeletal muscle aging is strongly affected by a loss of balance between damage and repair processes both at the molecular and the myocellular levels and is marked by immune activation [9]. The increase in chronic low-grade systemic inflammation during ageing is associated with sarcopenia and frailty and involves an increase in resident macrophage populations within aging muscles [10]. M2 macrophages were reported to be elevated in aged mouse skeletal muscle and associated with increased fibrosis. CCAAT enhancer-binding protein-α (C/EBPα) is a transcriptional regulator of adipogenesis, also highly expressed in macrophages and early myeloid progenitors, which regulates monocyte and macrophage development [11]. As a lymphocyte activation marker, lymphatic vascular endothelial hyaluronan receptor is associated with sarcopenia and systemic inflammatory response [12] and could be used as a novel protein biomarker for cancer cachexia [13]. However, information on the presence of macrophages in sarcopenic skeletal muscle is still sporadic and the effect of sarcopenia on inflammatory response remains unknown.

Low-magnitude high-frequency vibration (LMHFV) is a biophysical intervention proven to enhance muscle strength and balancing performance in community-dwelling older people [14]. Preclinical studies reported that LMHFV improved the structural and functional outcomes of the skeletal muscle [15] and prevented intramuscular fat infiltration in a sarcopenic mice model [16]. Magnesium (Mg) ion is an essential cofactor in a wide variety of biologic processes and dietary Mg deficiency is common in the elderly population [17]. Abundancy of dietary Mg was reported to associate with better indices of skeletal muscle mass and power in older adults, suggesting oral supplementation of Mg may be a potential strategy to tackle muscle atrophy [18,19].

The senescence-accelerated mouse prone (SAMP) model is characterized by accelerated ageing and a shortened life span. Our laboratory has established a sarcopenia animal model in the senescence-accelerated mouse prone-8 (SAMP8) that would present sarcopenic phenotypes at the age of 10 months [15,20]. In this study, we hypothesized that (1) oral supplementation of Mg or LMHFV could attenuate degeneration of muscle structure, function and strength in a sarcopenic senescence-accelerated mouse-P8 (SAMP8) mice model; (2) Mg or LMHFV could increase muscle hypertrophy and reduce inflammation, thus suppressing muscle atrophy during sarcopenia; (3) combined application of LMHFV and Mg could lead to a synergistic effect (4) through the PI3K/Akt/mTOR signaling pathway.

## 2. Results

### 2.1. In Vivo Results

#### 2.1.1. VIB and Mg Treatments Could Enhance Whole-Body Composition and Fiber Type Switch during Sarcopenia

The VIB treatment, Mg treatment and combination treatment started on 6-month-old SAMP8 mice and samples were collected when mice were 8 months old (month 2), 9 months old (month 3) and 10 months old (month 4). SAMP8 mice in the Con group showed a decreasing trend of serum Mg levels from month 2 to month 4 (Figure 1A), and the Mg group showed significantly increased levels compared to the Con group at month 3 and 4 (*p* < 0.01 and *p* < 0.001 respectively).

VIB treatment alone showed enhancements on whole body muscle mass/percentage and appendicular muscle mass/percentage at month 3 post treatment (Figure 1B–F). Lean mass and appendicular lean mass in VIB group were 6.00% and 15.10% higher than in the Con group at month 3 (both at *p* < 0.05). The VIB group continued to present higher percentage lean mass and percentage appendicular lean mass at month 4 (both at *p* < 0.01). The cross-sectional area (CSA) of type IIa muscle fiber in the VIB group was significantly higher than in the Con group (*p* < 0.05) at month 3. The percentage of type IIa fiber in the VIB group was higher than in the Con group (53.52%), Mg group (46.19%) and combination group (59.12%) at month 3 (*p* < 0.0001, *p* < 0.01, and *p* < 0.0001, respectively, Figure 2C). The percentage of type I fiber in the VIB group was higher than in the Con group (57.48%), Mg group (49.61%) and Mg + VIB group (60.36%) at month 3 (*p* < 0.0001, *p* < 0.01, and *p* < 0.0001, respectively). 

For whole body composition, the Mg group showed significantly higher lean mass and lean mass percentage at month 4 (both at *p* < 0.05, Figure 1C,D). More results of the wet weight were shown in Appendix A. The CSA of type IIb muscle fiber in the Mg group was higher than in the Con group (82.19%, *p* < 0.0001), VIB group (55.81%, *p* < 0.001) and Mg + VIB group (35.92%, *p* < 0.05) (Figure 2D), while the CSA of type I muscle fiber in the Con group was lower than in the Mg group (70.05%, *p* < 0.0001), VIB group (53.37%, *p* < 0.05) and Mg + VIB group (58.84%, *p* < 0.01). The CSA of Type IIa muscle fiber in the Con group was lower than in the Mg group (67.10%, *p* < 0.001), VIB group (62.27%, *p* < 0.01) and Mg + VIB group (73.32%, *p* < 0.0001) at month 4 (Figure 2D). Mg group showed the lowest percentage of type I fiber, the lowest percentage of type IIa fiber and the highest percentage of type IIb fiber compared with the Con group (66.68%, *p* < 0.0001; 46.31%, *p* < 0.0001 and 43.95%, *p* < 0.001, respectively) at month 4 (Figure 2D).

#### 2.1.2. Combined Treatment Yielded Better Treatment Effects in Whole Body Composition and Modulated Muscle Fiber Type Development during Progress of Sarcopenia

At month 2 post treatment, the Mg + VIB group showed higher lean mass (8.72%, *p* < 0.01) and appendicular lean mass (18.48%, *p* < 0.05) compared with the Con group at month 2 (Figure 1C,E), while the lean mass percentage and appendicular lean mass percentage in the Mg + VIB group were higher than in the Con group (both at *p* < 0.01, Figure 1D,F). The CSA of type I and IIa muscle fiber in the Mg + VIB group was higher than in the Con group (39.57%, *p* < 0.05; 59.73%, *p* < 0.0001) at month 2, and the CSA of type IIb muscle fiber in the Mg + VIB group was higher than in the Con group (72.95%, *p* < 0.0001), VIB group (40.09% *p* < 0.01) and Mg group (45.47%, *p* < 0.001) (Figure 2A,B). The Mg + VIB group showed a lower type I muscle fiber percentage than the Con group (66.03%, *p* < 0.0001) and Mg group (60.97%, *p* < 0.01) at month 2 and a higher type IIb muscle fiber percentage than the Con group (51.52%, *p* < 0.0001), VIB group (48.80%, *p* < 0.001) and Mg group (39.15%, *p* < 0.05) (Figure 2B). The Mg + VIB groups showed a significantly higher fiber percentage, with CSA between 1000–1500 μm^2^ and >1500 μm^2^, than the Con group at all time points (Appendix A).

#### 2.1.3. VIB and Mg Treatments Could Enhance Muscle Function to Ameliorate Muscle Aging Effects

In the ex vivo functional test, F_0_, SF_0_, F_t_, SF_t_ and contraction rate in the VIB group were higher than in the Con group at month 3 (*p* < 0.0001, *p* < 0.001, *p* < 0.001, *p* < 0.01, and *p* < 0.05 respectively) (Figure 3A–D and Appendix A). At month 4, the VIB group continued to show significantly higher fatigue force than the Con group at 10 s (*p* < 0.01), 500 s (*p* < 0.05) and 600 s (*p* < 0.05) (Figure 4F). More results of the ex vivo test were shown in Appendix A.

The Mg group showed the highest twitch and specific twitch force compared to the other three groups (both at *p* < 0.001 compared with Con group) at month 4 (Figure 3A,B). The tetanic force in the Mg group was higher than in the Con group (*p* < 0.001), VIB group (*p* < 0.05) and Mg + VIB group (*p* < 0.05) at month 4 (Figure 4C). The specific tetanic force in the Mg group was also higher than in the Con group (*p* < 0.0001), VIB group (*p* < 0.01) and Mg + VIB group (*p* < 0.05) at month 4 (Figure 3D). Only the Mg group showed a significantly higher contraction rate (*p* < 0.05) and shorter ½ relaxation time (*p* < 0.01) than the Con group at month 4 (Appendix A).

Handgrip strength was widely used to indicate skeletal muscle strength and function [21]. Both Mg and Mg + VIB treatments could increase grip strength at month 3 (36.87%, *p* < 0.0001 and 20.81%, *p* < 0.01, respectively), and grip strength in the Con group was lower than in the Mg + VIB group (27.77%, *p* < 0.001), Mg group (22.0%, *p* < 0.01) and VIB group (16.95%, *p* < 0.5) at month 4 (Figure 3E).

**Figure 3 ijms-23-12944-f003:**
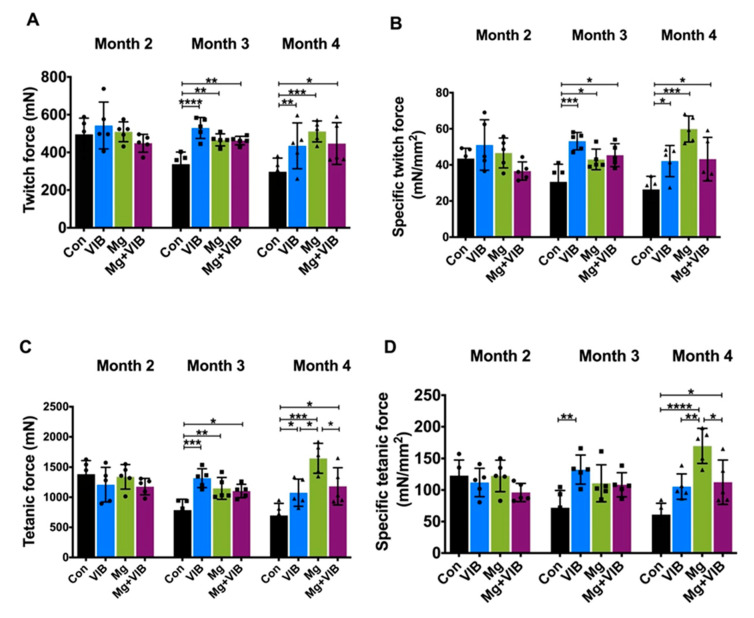
Muscle strength and function of SAMP8 mice upon different time points in treatment groups (n = 5). (**A**) twitch force, (**B**) specific twitch force, (**C**) tetanic force and (**D**) specific twitch force of VIB group were significantly higher than in all other three groups at month 3 post treatment, while the Mg group presented dominantly higher muscle twitch and tetanic contraction parameters at month 4. No significant difference was found at month 2. (**E**) Grip strength of the Mg group was significantly higher than that of the other groups at month 3, and the Mg + VIB group presented significantly higher grip strength at month 4. * *p* < 0.05, ** *p* < 0.01, *** *p* < 0.001 and **** *p* < 0.0001. (**F**) Mg treatment and combination treatment showed a higher fatigue curve and significantly higher fatigue force value than the Con group at every time point at month 4. The Mg group compared with Con group: **** *p* < 0.0001. The VIB group compared with the Con group: ▴ *p* < 0.05, ▴▴ *p* < 0.01 and the Mg + VIB group compared with the Con group: # *p* < 0.05, ## *p* < 0.01.

#### 2.1.4. Magnesium Supplementation and Vibration Treatment Regulated Macrophage Skewing and Reduced Inflammation during Sarcopenia

At month 10, the Mg group showed a significantly lower positive area of C/EBPα and LYVE1 than the Con group (both at *p* < 0.01) (Figure 4A,D,F), while the PAX7-positive cell numbers in Mg group were significantly higher than in the Con group (*p* < 0.01, Figure 4E,F) at month 10. VIB and Mg + VIB groups also showed a significantly lower LYVE1 positive area than the Con group (*p* < 0.01 and *p* < 0.5, respectively) (Figure 4D,F), with decreasing trends of the C/EBPα positive area. PAX7-positive cell numbers in the VIB group were significantly higher than in the Con group (*p* < 0.05, Figure 4E,F), while PAX7 -positive cell numbers in the Mg + VIB group were higher than in the Con group, without significance (*p* = 0.08). For macrophage polarization, CD206 positive areas in the VIB group, Mg group and Mg + VIB group were significantly lower than in the Con group (all at *p* < 0.0001, Figure 4B,F), and the F4/80 positive area showed no significant changes among all groups (Figure 4C,F).

#### 2.1.5. All Treatments Enhanced Muscle Proliferation and Prevented Expression of Muscle-Atrophy-Induced Ubiquitin Ligases via the PI3K/Akt/mTOR Pathway

The VIB, Mg and Mg + VIB treatments all showed positive effects on MRF expression at month 3 or month 4 (Figure 5A–C). At month 3, the MyoG and Myf6 mRNA expressions in the Mg + VIB group were higher than in the Con group (both at *p* < 0.05, Figure 5B). The expression of IGF-1 mRNA in the VIB group was higher than in the Con and Mg + VIB groups (both at *p* < 0.05) at month 3 (Figure 5B). The VIB group showed increased mTOR protein level (*p* < 0.05, Figure 5F) compared to the Con group and higher p70S6K and MyoD protein expression at month 3. No significant difference was observed in EIF4EBP1 protein expression among groups (Figure 5E).

At month 4, IGF-1 mRNA expression in the VIB group was higher than in the Con group (*p* < 0.0001), Mg group (*p* < 0.001) and Mg + VIB group (*p* < 0.0001) (Figure 5C), while the Mg group showed higher mRNA expression of MyoD (*p* < 0.01), MyoG (*p* < 0.05), Myf5 (*p* < 0.05) and Myf6 (*p* < 0.05) than the Con group (Figure 5C). At month 4, the FOXO3 mRNA expression level in the Con group was higher than in the Mg and Mg + VIB groups (both at *p* < 0.05). MuRF1 mRNA expression level in the Con group was higher than in the VIB group (*p* < 0.01) and Mg + VIB group (*p* < 0.05), and MAFbx mRNA expression level in the Con group was higher than in the VIB group (*p* < 0.001), Mg group (*p* < 0.001) and Mg + VIB group (*p* < 0.001) (Figure 5D).

The Mg + VIB group showed a dominant effect on pAkt activation and pAkt/Akt protein ratio at month 3 and 4 (Figure 5E,F). Expression of pAkt protein in the Mg + VIB group was higher than in the Con group (*p* < 0.01), VIB group (*p* < 0.05) and Mg group (*p* < 0.05) at month 3, while expression of pAkt protein in the Mg + VIB group was higher than in the Con group (*p* < 0.0001), VIB group (*p* < 0.001) and Mg group (*p* < 0.0001) at month 4 (Figure 5F). The pAkt/Akt ratio in the Mg + VIB group was higher than in the Con group (*p* < 0.01), VIB group (*p* < 0.05) and Mg group (*p* < 0.01) at month 4. Expression of mTOR protein in the VIB and Mg + VIB groups was higher than in the Con group (both at *p* < 0.05) at month 3, while expression of mTOR protein in the Mg group showed an increasing trend at month 3 and increasing effect compared to the Con group (*p* < 0.05) at month 4 (Figure 5F). The VIB and Mg + VIB groups showed an increasing effect on p70-S6K protein expression compared to the Con group, yet without significance at month 3 and 4, whereas no significant effect was observed on EIF4EBP1 among groups during sarcopenia.

**Figure 4 ijms-23-12944-f004:**
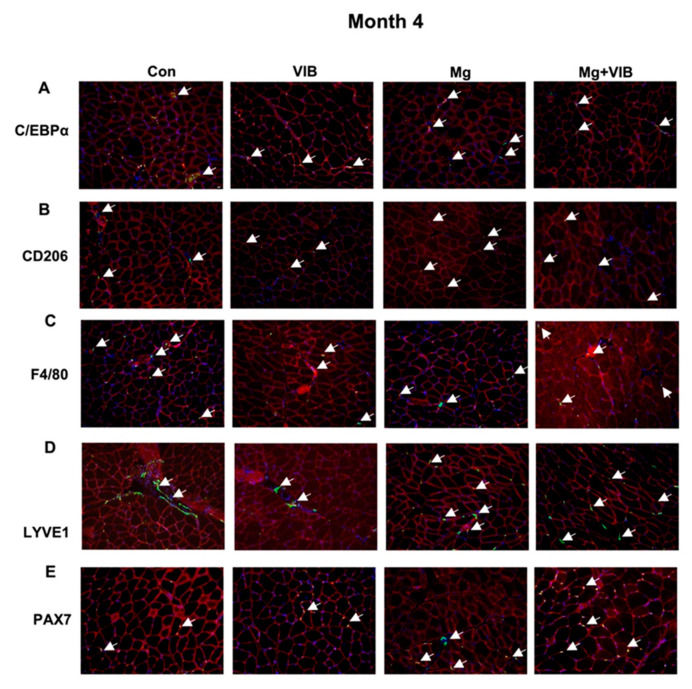
Immunohistochemical staining of inflammatory markers and M1/M2 macrophage in different groups at month 4 post treatment. White arrow showed the positive area. (**A**) At month 10, the Mg group showed a significantly lower positive area of C/EBPα staining area than the Con group (*p* < 0.01). The C/EBPα positive area in the VIB group and the Mg + VIB group were lower than in the Con group, without significance. (**B**) CD206 positive areas in all treatment groups were significantly lower than in the Con group. (**C**) The F4/80 positive area showed no significant changes among all groups. (**D**) VIB, Mg and Mg + VIB groups showed significantly lower LYVE1 positive areas than the Con group at month 10. (**E**) PAX7-positive cell numbers in the VIB and Mg groups were significantly higher than in the Con group. (**F**) Quantification results of C/EBPα, CD206, F4/80, LYVE1 positive area and PAX7-positive cell number. Scale Bar: 10 μm. * *p* < 0.05, ** *p* < 0.01, and **** *p* < 0.0001, n = 5–10.

**Figure 5 ijms-23-12944-f005:**
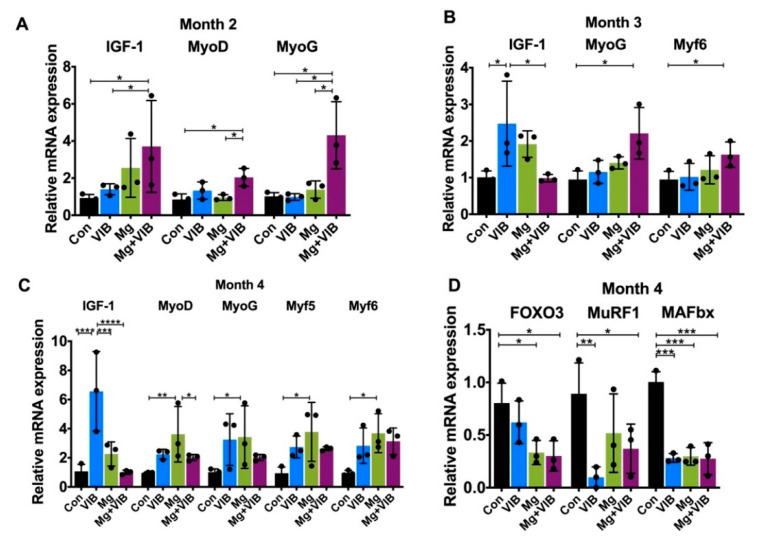
Treatment effects on muscle hypertrophy and atrophy in SAMP8 mice upon different time points. (**A**–**C**) Expression of IGF-1, MyoD, MyoG, Myf5 and Myf6 mRNA in different treatment groups at month 2, 3 and 4. (**D**) MuRF1 and MAFbx mRNA expression levels in the VIB group were significantly lower than in the Con group at month 4. (**E**,**F**) Western blots of protein expressions of mTOR, PI3K-p85, Akt, pAkt, p70-S6K, EIF4EBP1, MyoD and GAPDH at month 2, 3 and 4. * *p* < 0.05, ** *p* < 0.01, *** *p* < 0.001 and **** *p* < 0.0001, n = 3.

### 2.2. In Vitro Results

#### Myotube Formation of Treatment Groups via the PI3K/Akt/mTOR Pathway

To further confirm the role of the PI3K/Akt/mTOR pathway in the treatments, extra inhibition groups were added in vitro. The C2C12 cell viability was evaluated using MTT test (Appendix A). Figure 6A indicated that the myotubes of the VIB, Mg and combined treatment groups appeared morphologically distinct from the Con group. The myotube diameters of the VIB, Mg and Mg + VIB groups were larger than that of the Con group (*p* < 0.0001, *p* < 0.0001 and *p* < 0.0001 respectively) (Figure 6B). The myotube nuclei number of the VIB and Mg groups was higher than the Con group (*p* < 0.0001 and *p* < 0.001, respectively); The MI of the VIB and Mg groups were 2.734 times and 2.22 times higher than the Con group (*p* < 0.01 and *p* < 0.05, respectively) (Appendix A).

As shown in Figure 6C, the enhancing effect of VIB or Mg on myogenesis was diminished by the PI3K/Akt inhibitor and mTOR inhibitor. VIB + LY and VIB + Ra groups had a lower myotube diameter (both at *p* < 0.0001) and nuclei number (both at *p* < 0.0001) than the VIB group (Figure 6C,D). Both the Mg + LY and Mg + Ra groups had a significantly lower myotube diameter (both at *p* < 0.0001) (Figure 6C) and myotube nuclei number (both at *p* < 0.0001) (Figure 6D) than the Mg group. The M + V+LY and M + V+Ra groups had a lower myotube diameter (both at *p* < 0.0001) and nuclei number (*p* < 0.01 and *p* < 0.001, respectively) than the Mg + VIB group (Figure 6C,D). More results on MI are shown in Appendix A.

The VIB and Mg groups were able to enhance myoblast differentiation and significantly affect MHC type IIa expression via the PI3K/Akt/mTOR pathway (Figure 7A,D). Both the VIB and Mg groups had increased relative mRNA expression of Myf5 compared to the Con group (*p* < 0.01 and *p* < 0.05). The VIB and Mg + VIB groups showed a similar increasing effect on Myf6 mRNA expression compared with the Con group (*p* < 0.001 and *p* < 0.05) (Figure 7B). VIB treatment increased relative protein expressions of p85 (*p* < 0.05), Akt (*p* < 0.05) and mTOR (*p* < 0.05), while Mg treatment increased relative protein expressions of p85 (*p* < 0.01) and Akt (*p* < 0.05) (Figure 7E). Mg + VIB treatment significantly increased relative protein expressions of pAKT (*p* < 0.05) (Figure 7E).

After adding inhibitors, relative expression levels of Myf6 in the VIB + LY and VIB + Ra groups were lower than in the VIB group (both at *p* < 0.001) (Figure 7C). The Mg + LY and Mg + Ra groups showed decreased expression level of MyoG (*p* < 0.01 and *p* < 0.05) and Myf5 (*p* < 0.01 and *p* < 0.05) compared with the Mg group (Figure 7C). The relative mRNA expression levels of MyoG and Myf5 in the M + V+ LY groups were lower than in the Mg + VIB group (*p* < 0.05 and *p* < 0.001, respectively). The M + V+ Ra group had a decreased MyoG, Myf5 and Myf6 expression level compared with the Mg + VIB group (all at *p* < 0.05).

The VIB + LY and VIB + Ra treatments decreased the protein expression level of p85 (both at *p* < 0.05) and Akt (both at *p* < 0.05) compared with the VIB group (Figure 7E). The Mg + LY treatment decreased the levels of Akt (*p* < 0.01) and pAkt (*p* < 0.01), with a marginally significant difference in p85 protein expression compared with the Mg group (*p* = 0.06) (Figure 7E), while the Mg + Ra treatment significantly decreased the expression level of p85 (*p* < 0.05), Akt (*p* < 0.05) and pAkt (*p* < 0.05) compared with the Mg group. The M + V + LY and M + V+ Ra treatments decreased the protein expression level of p85 (both at *p* < 0.01) and pAkt (both at *p* < 0.01) compared with the Mg + VIB group. More western blot results of protein expressions of EIF4EBP1, phospho EIF4EBP1, p70-S6K and phospho p70-S6K in C2C12 were shown in Appendix A. 

**Figure 6 ijms-23-12944-f006:**
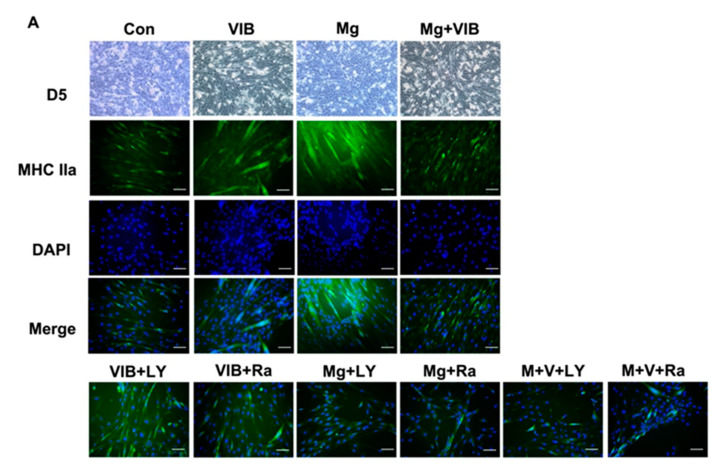
Myoblast differentiation and myotube formation in the C2C12 cell line with different treatments with PI3k/Akt inhibitor (LY294002) and mTOR inhibitor (Rapamycin). (**A**) Immunofluorescent staining of myotubes of VIB, Mg and combined treatment groups appeared morphologically distinct from the Con group. (**B**) Myotube diameters, myotube nuclei numbers of VIB, Mg and Mg + VIB groups showed a significant difference compared with the Con group. (**C**,**D**) Inhibitor groups showed significant smaller myotube diameters and myotube nuclei numbers. Scale Bar: 50 μm. * *p* < 0.05, ** *p* < 0.01, *** *p* < 0.001 and **** *p* < 0.0001, n = 10–15.

**Figure 7 ijms-23-12944-f007:**
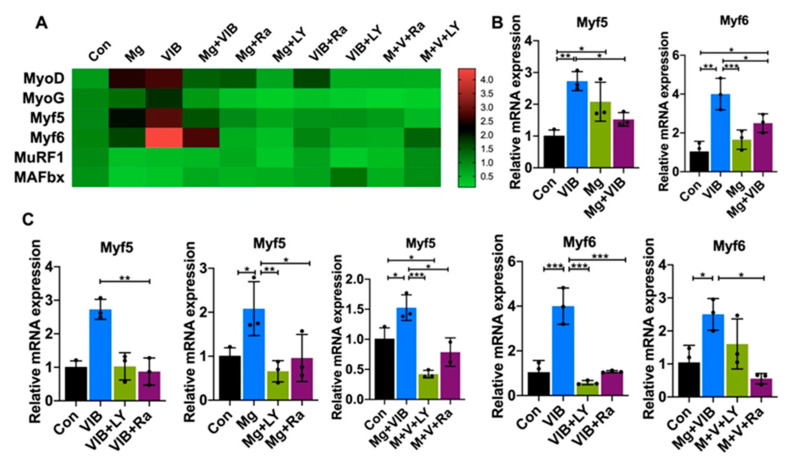
Treatment effects on myogenesis, atrogenes mRNA expression and PI3K/Akt/mTOR pathway with inhibitors in the C2C12 myoblast cell line. (**A**) Heatmap of MRF and atrogenes showed that (**B**) both the VIB and Mg groups could significantly increase the relative expression level of Myf5 mRNA compared with the Con group. The VIB and Mg + VIB groups showed a similar significantly increasing effect on Myf6 mRNA expression compared with the Con group. (**C**) Inhibitor groups had significantly decreased Myf5 and MyoG expression. (**D**,**E**) Western blots of protein expressions of mTOR, PI3K-p85, Akt, pAkt and GAPDH in different treatment groups in vitro. * *p* < 0.05, ** *p* < 0.01 and *** *p* < 0.001, n = 3–5.

## 3. Discussion

This study aims to investigate the combined effect of LMHFV and Mg in a sarcopenic animal model. In this study, the combination of Mg and LMHFV significantly increased lean mass, appendicular lean mass and fiber CSA among all groups during the progress of sarcopenia (month 2 post treatment). These results were similar to Yamada et al.’s study [22], which showed skeletal muscle mass index (SMI) and walking speed were significantly improved in the combination treatment of multinutrient supplementation and resistance exercise. Regarding muscle typing, the combination treatment showed the largest CSA of all fibers, the least proportion of type I fibers and the highest proportion of type IIb fibers during sarcopenia. Combination treatment of Mg and LMHFV could drastically enhance muscle mass (+8.7%) and alter muscle fiber type, confirming that the combination treatment could synergistically prevent the switch of fiber types during sarcopenia. Early studies suggested that muscle hypertrophy and fiber-type conversion were associated with alterations in MyoG, MyoD and IGF-I expressions in skeletal muscle [23]. In our study, positive correlations exist between muscle-fiber CSA and the expression of levels of IGF-I, MRFs, MyoG and MyoD. Levels of IGF-1 in combination treatment increased at different time points, whereas both MuRF1 and MAFbx were decreased at month 4, suggesting that upregulation of MRF could be a determining factor in the promotion of muscle mass. For muscle function, twitch force (F_0_) and grip strength were also significantly increased at month 4 in the combination group. Taking the results together, it is postulated that the combined treatment could improve muscle morphology first and then muscle function.

In the present study, we utilized a C2C12 myotube to examine whether the IGF-1/PI3K/Akt pathway affected the atrophy-induced upregulation of MuRF1 and MAFbx and hence the atrophy. Results demonstrated that the combination of LMHFV and Mg provided large synergistic effects on pAkt and mTOR protein expression. This underpins previous suggestions that the PI3K/Akt/mTOR pathway is one of the potential pathways by which resistance exercise and nutritional interventions may influence cellular events in the regulation of muscle mass during ageing [24]. Similarly, in vitro, the combination of LMHFV and Mg significantly increased C2C12 myoblast differentiation and myotube formation. Both Myf5 and Myf6 mRNA expression levels were significantly increased, consistent with LMHFV or Mg treatments individually. Inhibition of the Akt pathway suppressed treatment-induced MyoG expression, while MyoG and Myf5 expression could be inhibited by rapamycin, indicating Akt and mTOR pathways were both activated by the combination treatment of LMHFV and Mg.

For the individual treatments, LMHFV or Mg demonstrated beneficial effects to different extents on enhancements of muscle quality and control of muscle atrophy at different time points. The treatment effect of LMHFV on muscle fiber size and typing started at month 2. As type I muscle fibers were closely related to twitch, the increase in twitch and specific twitch forces by LMHFV was consistent with muscle fiber and mass changes. Tetanic parameters of the VIB group all peaked at month 3, which was consistent with our previous studies showing a positive influence on muscle mass, strength, fiber typing and myostatin suppression [15,16].

It has been reported that the proliferation of satellite cells could be affected by the niche of an inflammatory microenvironment [25]. This study found that myogenesis marked by PAX7-positive cells could be increased by Mg treatment and LMHFV treatment. A previous study has stated that M1 macrophages decline with ageing; the total number of macrophages were invariant with older age, with decreased M1 macrophage numbers and increased M2 numbers [26]. During sarcopenia, LMHFV or Mg could suppress muscle inflammation (C/EBPα and LYVE1) and modulate M2 macrophages in this study. The number of total macrophages (F4/80 positive cells) was found unchanged among all treatment groups, whereas M2 macrophage numbers were reduced, indicating that excessive inflammation and skewed macrophage phenotype during sarcopenia were attenuated by LMHFV treatment and Mg treatment.

A previous study demonstrated that decrease in muscle mass could be a result of decreased protein synthesis or increased degradation [27], particularly the ATP-dependent ubiquitin–proteasome proteolytic pathway [28,29]. LMHFV could inhibit upregulation of MAFbx and MuRF1 expression with increased pAkt/mTOR expression at different time points in SAMP8 mice. Previous studies showed that blocking mTOR function with rapamycin inhibited myotube hypertrophy and prevented both myogenic differentiation and myofibers growth in vitro [30] as well as muscle growth in vivo [31]. The blocking Akt function with LY294002 and the mTOR function with rapamycin could significantly inhibit VIB-induced myotube hypertrophy with lower p85 and pAkt protein expression.

Oral Mg supplement increased type II fibers but suppressed type I fibers. The Mg group showed a continuous, increasing trend of functional parameters during sarcopenia, including tetanic and specific tetanic force at month 4. A previous study showed that dietary Mg supplementation could improve intracellular ATP synthesis and thereby reduce mitochondrial ROS production in vascular smooth muscle cells of prematurely ageing Lmna^G609G/+^ [32]. These results confirmed that Mg was the most effective treatment on contractile functions versus LMHFV or the combination treatment. More prolonged use of Mg in doses that are higher than usual is needed to establish its routine or selective administration in patients with sarcopenia. In addition, the Mg supplement increased MyoD and Myf5 levels at month 4, with significantly suppressed MAFbx expression via increasing mTOR expression in vivo, indicating Mg treatment enhanced muscle proliferation in sarcopenic mice. The blocking Akt function with LY294002 and the mTOR function with rapamycin inhibited myotube hypertrophy and decreased Mg-induced Myf5 expression in C2C12 myoblasts, indicating that Mg could stimulate muscle growth by increasing MRF via the PI3K/Akt/mTOR pathway.

Although SAMP8 mice has been proven as a good animal model for sarcopenia [16,33,34], biomechanical variables (frequency, amplitude and knee angle) of muscle activity during LMHFV may differ between mice and humans.

In conclusion, combination treatment of LMHFV and Mg could synergistically suppress type I fiber atrophy but increase type II muscle hypertrophy during the progress of sarcopenia. It could improve muscle morphology first and then muscle function towards the later stage of sarcopenia. LMHFV showed a dominant effect on muscle mass and enhanced functions. An oral Mg supplement could attenuate deterioration of muscle function at the later stage of sarcopenia. Both treatments could modulate macrophage phenotype by decreasing the M2 population and reduce inflammation during sarcopenia. The PI3K/Akt/mTOR pathway was the predominant regulatory mechanism through which LMHFV and Mg enhanced muscle regeneration and suppressed atrophy. Coapplication of LMHFV and Mg can be considered as a novel intervention strategy against sarcopenia for future clinical studies and applications.

## 4. Methodology

### 4.1. Animal Model and Study Design

Male senescence-accelerated mouse prone 8 (SAMP8) mice were used as the sarcopenia animal model [15,20]. Sixty (n = 60) SAMP8 mice aged 6 months were randomized into the control (Con), LMHFV(VIB), magnesium (Mg) and combined treatment group (Mg + VIB). Mice in the VIB group were treated with LMHFV (35 Hz, 0.3 g, 20 min/day and 5 days/week) [20], whereas the Con group were given the same regimen with the platform powered off. Mg was administered to the animals through oral gavage at 0.2 mL MgCl_2_ solution in water (providing 200 mg/kg/day Mg ion, 5 days/week) [35,36]. Ad libitum regular chow formulated with 0.18% Mg were provided throughout the study period. Functional and structural outcomes were evaluated at endpoints of months 2, 3 and 4 post treatment (n = 5/group/time-point). All 4 groups shared the month 6 time point. The research protocol was approved by the Animal Experimentation Ethics Committee (AEEC) of the Chinese University of Hong Kong (Ref: 18-028-MIS).

At endpoints, gastrocnemius of right limb was harvested for ex vivo functional test while the contralateral muscle was harvested, weighted and snap-frozen with 2-methylbutane in optimal length for histological and immunofluorescence examination. Serum samples were collected for Mg concentration. The extensor digitorum longus (EDL) was collected for RNA extraction and quantitative real-time PCR, and Tibialis anterior (TA) for Western blot analysis.

### 4.2. Grip Strength Measurement

Forelimb grip strength of mice was measured with a force gauge (Mark-10 Corporation, Copiague, NY, USA). Mice held by the tail grasped the grid connected to the force gauge with their forepaws. The tails of mice were pulled slowly until the mice released their forepaws from the grid [16]. The peak force of each test was recorded with three repeats and averaged.

### 4.3. Muscle Mass by Dual-Energy X-ray Absorptiometry (DXA)

DXA (UltraFocus DXA, Faxitron, Tucson, AZ, USA) was used to quantify whole-body muscle mass and appendicular muscle mass at month 0, 2, 3 and 4 post treatment before euthanasia. Mice were placed in prone position with four limbs fixed. Imaging scans were taken and whole-body composition was analyzed by the BiopticsVision software. Appendicular lean mass was manually identified from the knee joint to the ankle joint for analysis.

### 4.4. Muscle Strength by Ex Vivo Functional Test

The ex vivo functional test (800A, Aurora Scientific, Aurora, ON, Canada) was conducted as previously described [15,16,20,34]. Muscle was electronically stimulated at optimal length by a single stimulus to evaluate twitch force (F0), specific twitch force (SF0), tetanic force (Ft), and specific tetanic force (SFt) at 150 Hz for 300 ms. Fatigue and MCSA were analyzed by the Dynamic Muscle Analysis system (DMA v5.3.2) as previously described [15].

### 4.5. Serum Mg Levels

Serum Mg concentration was determined with the Magnesium Assay Kit (ab102506, abcam, Cambridge, UK) [37]. Briefly, 50 μL serum samples and dilutions of a standard Mg solution were mixed with detecting reagents in microplates and incubated for 10 min at 37 °C and read at 450 nm. Mg concentration was expressed per 50 μL sample volume.

### 4.6. C2C12 Myoblast Cell Culture

C2C12 myoblast cells were maintained in Dulbecco’s modified minimum essential medium (DMEM) supplemented with 10% fetal bovine serum (Gibco, Grand Island, NY, USA), 3.7 g/L sodium bicarbonate (Sigma Aldrich, St. Louis, MO, USA) and 1% Penicillin-Streptomycin-Neomycin (PSN) Antibiotic Mixture (Gibco) at 37 °C in air with humidified atmosphere of 5% CO_2_ [20]. The myoblasts were fused into myotubes at 80% confluence by differentiation medium consisting of DMEM supplemented with non-heat inactivated 2% horse serum. There were 10 groups: (1) control (Con), (2) LMHFV (VIB), (3) Mg, (4) Mg + LMHFV (Mg + VIB), (5) LMHFV + Rapamycin (VIB + Ra), (6) LMHFV + LY294002 (VIB + LY), (7) Mg + Rapamycin (Mg + Ra), (8) Mg + LY294002 (Mg + LY), (9) Mg + LMHFV + Rapamycin (M + V + Ra) and (10) Mg + LMHFV + LY29402 (M + V+ LY). The time point at which differentiation was induced was regarded as day 0 (D0). A total of 10 mM Mg and LMHFV (35 Hz, 0.3 g; 20 min/day) were administered on D1; PI3K/Akt inhibitor (20 μM LY294002) and mTOR inhibitor (50 nm Rapamycin) were added on D3. Myotube formation was examined by immunofluorescence on D5. 

### 4.7. Immunohistochemical and Immunofluorescence Staining of Myofibers and C2C12 Myotubes

Transverse cryosections obtained from the left gastrocnemius (middle belly) were cut at 8 μm. Immunohistochemistry of MHC was performed based on our previous protocol [16]. Primary antibodies against mouse MHC I (BA-F8, blue, 1:200, Developmental Studies Hybridoma Bank, Iowa, IA, USA), MHC IIa (SC-71, green, 1:200, Developmental Studies Hybridoma Bank, Iowa, IA, USA) and MHC IIb (BF-F3, red, 1:200, Developmental Studies Hybridoma Bank, Iowa, IA, USA) were mixed at 4 µg/mL into a cocktail. Secondary antibody cocktail (Alexa Fluor 350 IgG2b, A-21140; Alexa Fluor 488 IgG1, A-21121; Alexa Fluor 555 IgM, A-21426, Thermo scientific, Waltham, MA, USA) were mixed at 4 µg/mL and incubated for 60 min. Immunostaining of C/EBPα (green, 1:200, ab40761, abcam, Cambridge, UK), CD206 (green, 1:200, ab64693, abcam, Cambridge, UK), F4/80 (green, 1:200, 30325, Cell signaling Technology, Danvers, MA, USA), LYVE1 (green, 1:100, ab14917, abcam, Cambridge, UK), PAX7 (green, 1:200, Developmental Studies Hybridoma Bank, Iowa, IA, USA) and Laminin (red, 1:200, ab11576, abcam, Cambridge, UK) was also performed. Primary antibodies were incubated overnight at 4 °C. Secondary antibodies were goat anti-rabbit IgG (H + L), Alexa Fluor 488 (1:200, a11034, Thermo scientific, Waltham, MA, USA) and goat anti-rat IgG H&L preabsorbed (1:200, ab7094, abcam, Cambridge, UK). Slides were visualized under a fluorescence microscope (Leica, DM 6000B, Werzlar, Germany) at 200× magnifications and analyzed with Image J (U. S. National Institutes of Health, Bethesda, MD, USA).

In vitro, C2C12-derived myotubes were stained with antibodies against MHC IIa and secondary antibody Alexa Fluor 488 IgG1 at 4 µg/mL. Nuclei were counterstained with DAPI as recommended by the manufacturer [38]. Myotube metrics were quantified using ImageJ software to determine myotube diameter and myotube nuclei numbers. A myogenic index (MI) was also calculated to indicate myotube fusion, and the mean number of nuclei per myotube was calculated to indicate heterogeneity of myotube size [39,40].

### 4.8. MRF and Atrogene mRNA Expression by Real-Time PCR

Total RNA isolated from EDL and C2C12 cells were harvested with RNAiso Plus reagent (TaKaRa, Shiga, Japan) following the manufacturer’s protocol. After extraction, cDNAs were reverse-transcribed from RNA using PrimeScript RT Master Mix kit (TaKaRa, Shiga, Japan). Power SYBR Green PCR Master Mix (Thermo scientific, Waltham, MA, USA) was used to perform the qRT-PCR of target mRNA detection (ABI 7300, Applied Biosystems, Waltham, MA, USA) based on our protocol [15,16]. Relative fold changes of candidate genes were analyzed by the 2−ΔΔCt method. Primers used for qRT-PCR are shown in Table 1.

### 4.9. Protein Expression by Western Blot

TA muscle samples and C2C12 cells were homogenized and lysed in radioimmunoprecipitation assay (RIPA) buffer with complete mini protease/phosphatase inhibitor cocktail (5872S, Cell signaling Technology, Danvers, MA, USA) according to the manufacturer’s instructions. Western blot analysis was performed according to our previous protocol [15]. Primary antibodies used were GAPDH (1:2000, MA5-15738, Invitrogen, Waltham, MA, USA), PI3 Kinase p85 (1:2000, 4292, Cell signaling Technology, Danvers, MA, USA), Akt (1:2000, 4691, Cell signaling Technology, Danvers, MA, USA), pAkt (1:2000, 4060, Cell signaling Technology, Danvers, MA, USA), mTOR (1:2000, 5043, Cell signaling Technology, Danvers, MA, USA), p70-S6K (1:2000, 34475, Cell signaling Technology, Danvers, MA, USA), EIF4EBP1 (1:2000, 9644, Cell signaling Technology, Danvers, MA, USA) and MyoD (1:2000, 13812, Cell signaling Technology, Danvers, MA, USA). Secondary antibodies used were anti-rabbit IgG, HRP-linked antibody (1:5000, Cell signaling Technology, Danvers, MA, USA) and goat anti-mouse IgG (H + L) antibody (1:5000, Invitrogen, Waltham, MA, USA). Relative protein contents were imaged by GeneGnome XRQ (Syngene, Cambridge, UK) and quantified by ImageJ.

### 4.10. Statistical Analysis

All quantitative data were expressed as means ± standard deviation. One-way ANOVA and post hoc Tukey’s test were used for comparison of different groups at the same time points (Prism8, GraphPad Software, San Diego, CA, USA). Statistical significance was set at *p* < 0.05.

## Figures and Tables

**Figure 1 ijms-23-12944-f001:**
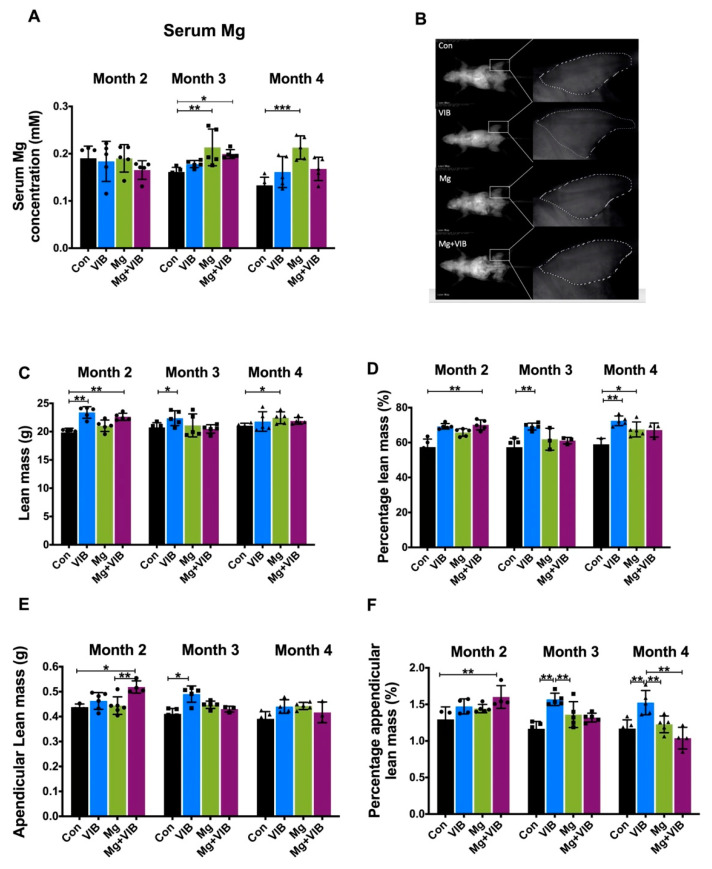
Body mass parameters by Dual Energy X-rat Absorptiometry (DXA) and serum Mg concentration of SAMP8 mice. (**A**) Serum Mg levels in Con group showed a decreasing trend from month 2 to month 4, and serum Mg concentration in the Mg group was significantly higher than in the Con group at month 4. (**B**) Representative DXA images in lean mass mode. (**C**) Lean mass, (**D**) lean mass percentage, (**E**) appendicular lean mass and (**F**) appendicular lean mass percentage of SAMP8 mice in different groups at month 2, 3 and 4 post treatment. Lean mass in the Mg group was significantly higher than in the Con group at month 4. Mg + VIB group showed a significantly higher lean mass/percentage and appendicular lean mass/percentage compared with the Con group at month 2, and the VIB group showed significantly higher mass/percentage and appendicular lean mass/percentage compared with the Con group at month 3 and significantly higher lean mass percentage and percentage lean mass at month 4. * *p* < 0.05, ** *p* < 0.01, *** *p* < 0.001, n = 3–5.

**Figure 2 ijms-23-12944-f002:**
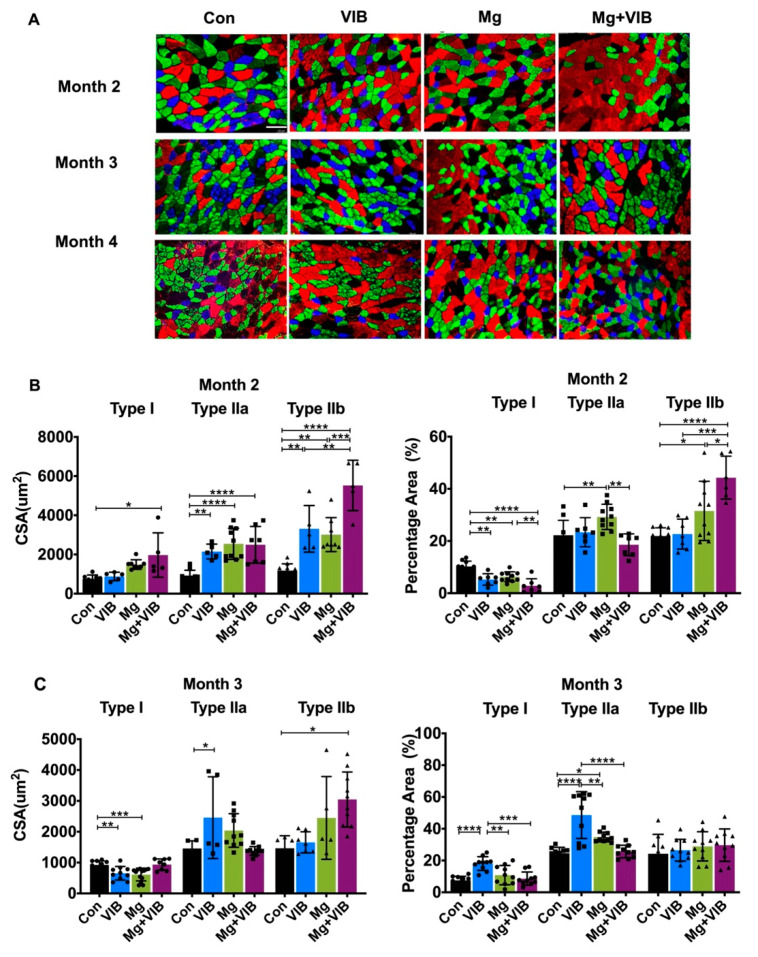
Immunohistochemical staining of myofibers (myosin heavy chain expression) in different groups at month 2, 3 and 4 post treatment. (**A**) Type I muscle fibers (blue), type IIa muscle fibers (green) and type IIb muscle fibers (red) of gastrocnemius at month 2, 3 and 4. (**B**–**D**) Type I, IIa and IIb CSA and percentage area in different treatment groups at month 2, 3 and 4. Scale Bar: 100 μm. * *p* < 0.05, ** *p* < 0.01, *** *p* < 0.001 and **** *p* < 0.0001, n = 5–10.

**Table 1 ijms-23-12944-t001:** Primer sequence for real-time PCR.

Gene	Sequence (5′to 3′)
MyoD	Forward: CCACTCCGGGACATAGACTTG Reverse: AAAAGCGCAGGTCTGGTGAG
MyoG	Forward: GAGACATCCCCCTATTTCTACCA Reverse: GCTCAGTCCGCTCATAGCC
Myf5	Forward: CACCACCAACCCTAACCAGAG Reverse: AGGCTGTAATAGTTCTCCACCTG
Myf6	Forward: AGAGGGCTCTCCTTTGTATCC Reverse: CTGCTTTCCGACGATCTGTGG
MAFbx	Forward: CAGCTTCGTGAGCGACCTC Reverse: GGCAGTCGAGAAGTCCAGTC
MuRF1	Forward: CCAGGCTGCGAATCCCTAC Reverse: ATTTTCTCGTCTTCGTGTTCCTT
FOXO3	Forward: CTGGGGGAACCTGTCCTATG Reverse: TCATTCTGAACGCGCATGAAG
IGF-1	Forward: GCTCTTCAGTTCGTGTGTG Reverse: CCTCAGATCACAGCTCCGGAAG
GAPDH	Forward: AACGACCCCTTCATTGAC Reverse: TCCACGACATACTCAGCAC

## Data Availability

The datasets generated during and/or analyzed during this study are available from the corresponding author on reasonable request.

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
