# Peer review of "Coapplication of Magnesium Supplementation and Vibration Modulate Macrophage Polarization to Attenuate Sarcopenic Muscle Atrophy through PI3K/Akt/mTOR Signaling Pathway"

_ijms, 2022, doi:10.3390/ijms232112944_

Round 1

Reviewer 1 Report

The manuscript structure and text do not allow a clear workflow understanding. The results start with Mg+VIB combination without any further explanation of single treatments neither the rationale of the experiments. The text is too descriptive always indicating unnecessarily the p values. In the 2.1 paragraph the data refer to random figures skipping different panels (Fig 1,2 5 ..). Fig1 A is lacking as well as panels in 2A. Is difficult to follow the text.

Reviewer 2 Report

Cui et al. provide an extensive study concerning effects of magnesium and vibration on the development of sarcopenia. To address this issue, they used a combination of in vivo and in vitro approaches, while combining different morphological, functional, and biochemical analyses.

Comments:

  1. Introducuction:
  • »Recent findings showed that activation of PI3K/Akt/mTOR pathway was important as a response to mechanical stimulation in muscle hypertrophy [4]« Original studies describing the role of mTOR in muscle hypertrophy are not “recent” they go back 10 years and more.
  • The introduction should, at least initially, make a distinction between sarcopenia, which can occur in young or old people, and ageing-related sarcopenia.
  • The multifactorial etiopathogenesis of sarcopenia should be highlighted in the introduction. For instance, ageing-related loss of motor neurons, which is one of major factors leading to sarcopenia, should at least be mentioned.
  • The sarcopenic model used in this study (SAMP8 mice) should be described in the introduction. The mechanisms of sarcopenia in this model should also be mentioned. Previous studies using the model should be referred to.

  1. Results and methods:
  • The authors should rethink the way they report the data. As written, it is rather difficult to follow the text. For instance, data presented in Figure 4 are mentioned in the text (in section 2.2) before data in Figure 3 (in section 2.3) etc. After reporting the Figure 3 data, the text goes back to Figure 1 and Figure 4 (in section 2.4) before discussing Figure 2 data once more in section 2.5 together with the Figure 5. On the other hand, Figure 5 is first mentioned already in section 2.1. The authors should carefully reconsider the sequence of Figures and/or organization of the text.
  • A simplified and a more systematic approach would significantly benefit the manuscript. Also, the authors should make sure that the same topic (e.g. fiber type composition, the size of fibers), which is reported in a single figure, is not spread over several sections, interrupted by description of data from other figures, while also making sure that the same results are not reported/discussed twice in different sections.

  • Figure legends should include n values. It is important to know how many mice, experiments etc. were done to obtain data. Also, especially for the in vivo part, it would be much better to uses bar charts together with scatter plot to indicate individual measurements.
  • Blots: the position of the molecular weight markers (that were used in this study) should be included in all pictures of the blots.
  • Some figure legends are very long (300 words or more).
  • Experimental treatments of C2C12 cell cultures should be described more clearly and in more detail.
  • Different muscles (gastrocnemius, EDL, tibialis anterior) were used for different analyses. However, these muscles do not have the same fiber type composition. Could this affect the interpretation of the data? Also, it would be important to know which part of the gastrocnemius muscle was used for analyses.
  • Since this is a sarcopenia study, the weight of the muscles should be reported.

  • Figure 2A: month 2 and month 4 data are missing.
  • Quantification of month 3 data (CSA) is missing in Figure 2.
  • The total fiber number is represented for type IIb fibers. Why not for other types as well?
  • Did authors assess viability of C2C12 cells after VIB?
  • Authors should check the numbering of Figures in section 2.7. E.g.: results of Akt are not presented in Figure 6.
  • Figure 7: It would be important to show that rapamycin really suppressed mTOR signalling. For instance, phosphorylation of 4E-BP1 and p70S6K should be reported in C2C12 cells treated with rapamycin.
  • Figures 6 and 7: Two important controls groups are missing: 1) C2C12 cells treated with rapamycin alone and 2) C2C12 cells treated with LY294002 alone. To fully appreciate the data and provide a correct interpretation these two groups are absolutely essential.

Reviewer 3 Report

The manuscript ijms-1682742 is interesting and aimed to demonstrate how magnesium supplementation together with an LMHFV protocol is able to attenuate sarcopenic muscle atrophy by acting on PI3K / Akt / mTOR signaling.

The manuscript takes into consideration an in vivo experimental protocol on a sarcopenic mouse model and an in vitro model with C2C12 myotubes. And this certainly gives value to the manuscript.

However, there are some critical issues that do not make this manuscript publishable in this form on IJMS.

a) First of all, the results are difficult to follow. In discussing the results obtained, we move from Figure 1 to 2 and then to 5. Next we comment on Figure 4 and after 3. It is downright confusing.

b) In paragraph 2.1 perhaps we should define at what age of the mice the protocol started, how long it lasted and comment that the analyzes were done 2-3-4 months after the end of the protocol.

c) Figures 1 and 2 are missing graphics / figures.

d) In some cases, as in Figure 3, the histograms are too small. They can only be deciphered by reading the article by amplifying the magnification.

e) In the materials and methods the authors describe grip strength protocol, but then it is not found in the figures / results.

f) In figure 5 there is no correspondence between histograms and images of the WB. I also don't understand the choice of which data to show in the histograms. Has a selection been made?

g) In the in vitro studies section, the C2C12 model is not mentioned except in the captions of the figures. Then it is not clear whether we are evaluating the differentiation process, myoblasts or myotubes. The comments highligt results on myotubes but throught the manuscript the comment is on about the differentiation process. It is not quite the same.

h) for the in vitro atrophy study, was it not possible to use dexamethasone on myotubes?

i) check the x-axis values ​​of the histograms in figure 6. There seems to be something wrong.

l) under discussion the authors comment on early stage of sarcopenia. Is it the experimental model that qualifies it? Comment / clarify on this point.

m) Table 1: Since the authors have entered a comment for myogenic factors, a comment should also be entered for FOXO3, IGF-1 and GAPDH.

Round 2

Reviewer 2 Report

The revisions made by Cui et al. significantly improved the manuscript. There are only a few issues that need to be resolved:

-        The addition of EIF4EBP1 and pEIF4EBP1 results in Figure S6 is appreciated. However, rapamycin did not seem to have an effect on this protein. I strongly recommend to the authors to show another control that would demonstrate inhibition of mTOR by rapamycin. For instance, p70S6K or rpS6 would be especially useful and appropriate for this purpose.

-        The addition of individual measurements/data points (scatter) in some of the figures is highly appreciated. For the sake of consistency and transparency of data presentation, I recommend that individual data points/measurements are included in all graphs (at least in the main text if not also the supplement).

Minor

-        “phospho” is sometimes written as “phosphor” (probably due to autocorrection of the text by Word).

-        the unit should be corrected: “100 lg/ml streptomycin«

-        “u” is used for “micro” in different units throughout the manuscript. Shouldn’t the Greek letter μ be used?

Reviewer 3 Report

The authors responded sufficiently to the critical issues highlighted and the manuscript was modified accordingly.

Now it can be accepted for publication on IJMS.

Author Response

Thank you for your comments, English language and style have been checked. 

Round 3

Reviewer 2 Report

Revisions by the authors are appreciated. I have no further comments.

Author Response

(The authors gave the same response as above.)
